Methods

# Spatial metabolomics for symbiotic marine invertebrates

Wing Yan Chan[1,2,*] , David Rudd[3,4,*], Madeleine JH van Oppen[1,2]

**Microbial symbionts frequently localize within specific body structures or cell types of their multicellular hosts. This spatiotemporal niche is critical to host health, nutrient exchange, and fitness. Measuring host–microbe metabolite exchange has conventionally relied on tissue homogenates, eliminating dimensionality and dampening analytical sensitivity. We have developed a mass spectrometry imaging workflow for a soft- and hard-bodied cnidarian animal capable of revealing the host and symbiont metabolome in situ, without the need for a priori isotopic labelling or skeleton decalcification. The mass spectrometry imaging method provides critical functional insights that cannot be gleaned from bulk tissue analyses or other presently available spatial methods. We show that cnidarian hosts may regulate microalgal symbiont acquisition and rejection through specific ceramides distributed throughout the tissue lining the gastrovascular cavity. The distribution pattern of betaine lipids showed that once resident, symbionts primarily reside in light-exposed tentacles to generate photosynthate. Spatial patterns of these metabolites also revealed that symbiont identity can drive host metabolism.**

## Introduction

Microorganisms such as eukaryotic microalgae, bacteria, viruses, fungi, and archaea are the most abundant life forms on Earth. Their symbioses with multicellular hosts can improve host uptake of nutrients (McFall-Ngai et al, 2013), vitamins, and minerals; mitigate toxic compounds and control pathogens; play a critical role in host health (e.g., human gut microbiota); and form the basis of the ecological success of some of the most productive ecosystems on earth (e.g., coral reefs, tropical rainforests, deep-sea hydrothermal vents) (Peixoto et al, 2022). For example, the plant rhizosphere is inhabited by a wide range of bacteria, fungi, microalgae, viruses, and archaea (Mendes et al, 2013), which protect the host against biotic and abiotic stresses, promote growth, and increase nutrient availability (Chagas et al, 2018). Furthermore, in humans and crops,

microbiome dysbiosis is linked to severe chronic disease and stress (Peixoto et al, 2022).

Key symbionts of cnidarian animals, such as reef-building corals and sea anemones, are dinoflagellate microalgae in the family Symbiodiniaceae (Davy et al, 2012). These symbionts translocate products of carbon fixation and nitrogen assimilation to the host (Burriesci et al, 2012; Kopp et al, 2015a) and meet up to ~161% of the host's basal metabolic energy requirements (Muscatine et al, 1984). In exchange, they gain protection and host-derived inorganic nutrients (Davy et al, 2012). Metabolites play a central role in host–microbe interactions by mediating and controlling the basic processes of recognition, signaling, and communication (Song et al, 2015; Cleary et al, 2017; Chagas et al, 2018). In plants, for instance, lipids such as triacylglycerols, phospholipids, galactolipids, and sphingolipids are crucial for signaling and system-level energy storage (Bhattacharya, 2022).

To understand host–microbe interactions, tissue homogenates have traditionally been analyzed with liquid/gas chromatography coupled to mass spectrometry (LC/GC–MS) to measure the host/symbiont metabolite profiles. However, symbiotic microorganisms are often found within specific anatomical structures or cell types of the host. For instance, the sulfur-oxidizing and methane-oxidizing gammaproteobacterial symbionts of deep-sea mussels (*Bathymodiolus puteoserpentis*) from hydrothermal vents are only found in the organism's epithelial cells (Geier et al, 2020). Similarly, the bioluminescent bacterium *Vibrio fischeri* is localized within the epithelial layers in the light organ of the Hawaiian bobtail squid (*Euprymna scolopes*) (Cleary et al, 2017). Spatial information is lost in tissue homogenates and the sensitivity for measuring specific foci of nutrient/metabolite exchange (e.g., within a particular body structure) is dampened in bulk analyses.

The loss of dimensionality substantially restricts the value of metabolomics to provide insights into the microbial ecology of symbioses as it can limit biological inferences to be drawn from the data. For instance, Williams et al (2021) used UHPLC-MS on tissue homogenates of the coral *Montipora capitata* in symbiosis with Symbiodiniaceae and detected an enrichment of montiporic acids under elevated temperature. Montiporic acids are cytotoxic and antimicrobial compounds found in *M. capitata* eggs that are also known to reduce the photochemical efficiency of coral microalgal symbionts (Hagedorn et al, 2015). Without knowing whether the

---

[1]School of BioSciences, University of Melbourne, Parkville, Australia   [2]Australian Institute of Marine Science, Townsville, Australia   [3]Monash Institute of Pharmaceutical Sciences, Parkville, Australia   [4]Melbourne Centre for Nanofabrication, Clayton, Australia

Correspondence: w.chan@unimelb.edu.au
*Wing Yan Chan and David Rudd contributed equally to this work.

enriched montiporic acids were co-localized with the gonads or the gastrodermal cells that contain the microalgal symbionts, it is impossible to tell if this result is linked to coral sexual reproduction or symbiont photosynthesis. The lack of spatial information can also reduce the ability to detect significant differences between experimental treatments. Sphingolipids can accumulate in host microalgal symbiont-containing gastrodermal cells of the sea anemone *Exaiptasia diaphana*. However, contrary to expectations, LC–MS/MS analysis of homogenized tissues did not reveal a significant sphingolipid concentration difference between anemones exposed to different light/dark treatments, possibly because of the dilution effect of non-symbiotic cells (Kitchen et al, 2017). These examples highlight that spatial knowledge of molecular species is key for obtaining in-depth understanding of host–microbe interactions.

Nanoscale secondary ion mass spectrometry (NanoSIMS) is one technique to study host–microbe interactions spatially. NanoSIMS is an isotope and elemental imaging technique that, in combination with transmission electron microscopy, can visualize relative isotopic enrichment in biological samples at very high spatial resolution (up to 50 nm) (Nuñez et al, 2018; Siegel et al, 2018). Through $^{13}$C and $^{15}$N labelling, NanoSIMS has enabled researchers to explore processes such as symbiont carbon assimilation and translocation (e.g., Ros et al, 2021) and symbiont nitrogen assimilation rate and nutrient competition in situ in corals (e.g., Krueger et al, 2020). However, the high energy monoatomic beam of NanoSIMS results in severe fragmentation of larger sized molecules (e.g., proteins, lipids) and limits researchers to study very small molecules only (i.e., < 200 m/z), and the technique can only image seven molecules at once because of its mass analyzer limit (Siegel et al, 2018). Furthermore, NanoSIMS requires samples with a skeleton (e.g., scleractinian corals) to be decalcified before imaging (Krueger et al, 2020; Ros et al, 2021) and hence spatial information across the coral polyp is poorly preserved.

MSI metabolomics, recently termed spatial metabolomics (Alexandrov, 2020), is one method to overcome the limitations of traditional metabolomics and NanoSIMS. MSI uses intact tissue sections, allowing researchers to simultaneously examine the spatial distribution of hundreds to thousands of molecules in complex biological samples in situ (Boughton & Hamilton, 2017). Unlike NanoSIMS, isotopic labelling or endoskeleton decalcification is not required for MSI. Through 2D rasterized ionization, the method generates thousands of charged species that are introduced into a mass spectrometer, capable of detecting a wide m/z range of 20–500 k+. Mass-to-charge (m/z) values detected on each position of the 2D raster pattern can then be related to molecular species. These positional spectra (termed pixels) are summed across the whole 2D area, where ion density maps can be generated to visualize the spatial location and relative intensity of each metabolite (Siegel et al, 2018). Although the metabolite peak area itself does not reflect absolute concentrations (as this depends on, e.g., ionization efficiency, transmission efficiency through the mass spectrometry), the peak area scales linearly with metabolite concentration and comparison of relative intensity between samples can be made (Liu & Locasale, 2017). Furthermore, absolute quantitation can be achieved through the use of external/internal standards (Unsihuay et al, 2021).

The spatial information provided by MSI has been proven highly valuable in medical studies, particularly for drug discovery, disposition, and disease state assessment (Boughton & Hamilton, 2017; Siegel et al, 2018; Roberts et al, 2022) and human–cancer cell, human–parasite, and human–bacterial pathogen interactions (Kloehn et al, 2021; Neumann et al, 2021; O'Neill et al, 2022). Sea anemones are rich in peptides with unusual pharmacological and structural properties; hence, they have been a target of venom studies for drug discovery (Mitchell et al, 2017; Madio et al, 2018; Surm et al, 2019). These MSI studies focused on the peptide mass range (usually m/z > 1,000) and have revealed that venoms were exclusively localized in anemone tentacles, likely linked to their role in prey capture (Mitchell et al, 2017; Madio et al, 2018).

Although MSI is well integrated into medical studies and other biological research such as on plants (Gupta et al, 2019; Nakabayashi et al, 2019) and invertebrates (Kopp et al, 2015b; Yang et al, 2020; Bourceau et al, 2022; Hamilton et al, 2022), only a few non-medical studies have explored its application in host-symbiont interactions. The ability to study hundreds to thousands of metabolites in situ via MSI makes this a powerful tool for researchers to decipher host–microbe interactions. Examples are the mapping of the spatial metabolome of a deep-sea mussel which has intracellular bacteria embedded within epithelial cells (Geier et al, 2020); investigating the role of bacteria-derived secondary metabolites in larval maturation of a marine snail (Rudd et al, 2015); and bacteria-derived prophylaxis in wasp cocoons (Kroiss et al, 2010). MSI has provided invaluable insights and new discoveries in these studies, such as the discovery of specialized metabolites at the host–microbe interface specific for mussel-methane–oxidizing bacterial interactions (Geier et al, 2020).

MSI has the potential for elucidating the cnidarian-dinoflagellate symbiosis—an avenue yet to be fully explored. Coral bleaching is the loss of microalgal symbionts (Symbiodiniaceae) that often results in host mortality (Hughes et al, 2017, 2018). With < 2% of the Great Barrier Reef (GBR) having escaped bleaching since 1998 (Hughes et al, 2021), knowledge on host-symbiont interactions is critical to develop novel reef conservation and restoration interventions. Only one study has applied MSI on an anemone tentacle and found a distinct distribution pattern of three metabolites in the tenacle (Kopp et al, 2015b).

Obtaining intact full-body tissue sections to investigate cnidarian-Symbiodiniaceae interactions in situ with MSI is challenging. Typical cryosectioning techniques cannot maintain the natural shape of delicate soft-bodied cnidarians (e.g., sea anemones) nor are these methods able to cut through the $CaCO_3$ skeleton of hard-bodied cnidarians (e.g., scleractinian corals) without disrupting their tissue. We have overcome these challenges and present a spatial metabolomics workflow that can reveal host and Symbiodiniaceae metabolite profiles in situ at 50 $\mu$m resolution using intact tissue sections of a soft-bodied (the sea anemone, *E. diaphana*) and a hard-bodied cnidarian (the scleractinian coral, *Galaxea fascicularis*). To verify the biological application of the method, we test (1) if the metabolite distribution is linked to specific host anatomical structures and (2) if symbiont identity drives anemone host metabolism in a spatial manner.

# Results and Discussion

## A spatial metabolomic workflow for marine invertebrates

Spatial omics studies are rare in marine invertebrates (Geier et al, 2020; Goto-Inoue et al, 2020; Hamilton et al, 2022), and the difficulty in obtaining intact tissue sections in the organism's natural shape is a major hurdle. Through the combination of anaesthetizing, embedding, and cryofilm application, the cryosectioning workflow presented produced intact tissue sections for a soft-bodied (sea anemone *E. diaphana*) and a hard-bodied (coral *G. fascicularis*) cnidarian in their natural shape (Fig 1); this is a prerequisite for spatial metabolomics and other spatial analysis. This method is applicable to other soft- and hard-bodied marine invertebrates, opening up new opportunities for the broader marine invertebrate research field. All detected metabolites are immediately contextual in that they can be correlated to host/symbiont anatomical structures with known ecological functions. Compared with NanoSIMS, which is restricted to imaging seven ions at a time, a total of 631 metabolites (after background signal removal) were imaged in this study across the anemone–symbiont metabolome using MALDI-MSI (Supplemental Data 1). Of these, 208 (33%) were annotated, falling within 31 major groups covering structural, energetic, and signaling metabolites (Fig 2 and Table S1 and Supplemental Data 1). The remaining 67% represent ambiguous or unknown metabolites, alluding to a considerable "dark metabolome" in cnidarian tissues. Annotation in this space is still one of the major bottlenecks in MSI, but this study has provided the first step to fill this knowledge gap.

## MALDI-MSI reveals spatial distribution patterns of metabolites that link to functionality

### Tissues surrounding the gastrovascular cavity as a focal site of host-symbiont regulation

MSI revealed the localization of metabolites that would otherwise be lost in tissue homogenates. Among the 31 metabolite groups detected in anemones, ceramides (group Cer, GlcCer, HexCer; 26 of the major chemical species detected) were found to be spatially concentrated in the tissues surrounding the gastrovascular cavity and occasionally the column outer wall and actinopharynx of symbiotic anemones (Fig 3A–C). In contrast, betaine lipids (group DGCC, MGCC, DGTS; 13 of the major chemical species detected) were primarily concentrated in the tentacles of symbiotic anemones (Fig 3A–C). Interestingly, the ratio between these metabolite groups showed a clear pattern that separates anemones based on the identity of their Symbiodiniaceae symbiont (Fig 3C), indicating that symbiont identity can change system-level host metabolism (see section "Symbiont identity drives system-level host metabolism spatially"). A total of seven peaks with m/z > 1,000 (1,004.62, 1,018.6, 1,022.63, 1,025.60, 1,031.65, 1,049.66, 1,065.57) represent unannotated metabolites that were highly concentrated in the tentacles and are potential anemone venoms for future investigation (Fig S1).

The concentration of ceramides within the tissues surrounding the gastrovascular cavity reflects that it is the primary site of regulation of Symbiodiniaceae symbionts by the cnidarian host.

Ceramides are intermediates in the biosynthesis and metabolism of all sphingolipids—a class of lipids that are important constituents of biological membranes which function in cell signaling through the activation of specific G-protein-coupled receptors—regulating processes such as apoptosis, cell survival, inflammation, autophagy, and oxidative stress responses (Rosset et al, 2021; Bhattacharya, 2022). The sphingosine rheostat refers to the balance in cellular concentrations of pro-survival sphingolipids (i.e., sphingosine-1-phosphate; S1P) and pro-apoptotic sphingolipids (i.e., sphingosine and ceramide), which is a homeostatic process that determines cell fate (Kitchen et al, 2017; Rosset et al, 2021) (Fig 4). Pro-survival and pro-apoptotic sphingolipids are interconvertible by the catalytic activities of sphingosine kinase and S1P phosphatase (SGPP), where sphingosine kinase promotes the conversion to S1P and enhances cell survival and proliferation and SGPP promotes the conversion to sphingosine that activates pro-apoptotic cellular cascades (Fig 4). The regulatory mechanisms that balance sphingosine conversion could be features that determine resilience to and recovery from abiotic stress such as coral bleaching caused by ocean warming, which is a global concern for reef-building corals.

Although several previous studies have pointed towards a regulatory role of sphingolipids in cnidarian-Symbiodiniaceae symbiosis, none has explored their spatial distribution and associated implications. Anemones supplied with exogenous pro-survival sphingolipids (S1P) exhibited less bleaching under elevated temperatures compared with control anemones (Detournay & Weis, 2011), and the pro-apoptotic *SGPP* gene is down-regulated in symbiotic compared with aposymbiotic anemones (Rodriguez-Lanetty et al, 2006; Kitchen et al, 2017). Uptake of Symbiodiniaceae symbionts by cnidarian hosts occurs in the gastrovascular cavity, where they are incorporated into the host's gastrodermal cells via phagocytosis (Davy et al, 2012). A high concentration of ceramides in the tissues within the gastrovascular cavity supports the role of sphingolipids in cnidarian-Symbiodiniaceae symbiosis regulation and indicates that this is likely where the fate of the microalgal symbiont (i.e., being digested/expelled or incorporated as a symbiont) is determined by the sphingosine rheostat (Fig 4).

How the cnidarian sphingosine rheostat may be affected by the microalgal symbiont species or strain harboured has previously not been explored. This study demonstrates that the relative intensity of most of the ceramides is significantly higher in B1-anemones (with homologous symbionts) than C1-anemones (with heterologous symbionts) (Figs 3C and S2 and Table S2), indicating that symbiont identity can influence host ceramide levels. Given the potential role of ceramides in symbiont regulation, their relative intensity difference could suggest that B1- and C1-anemones were at different states of active symbiont recruitment. Most of the literature suggests that sea anemones hosting homologous Symbiodiniaceae can achieve faster cell density recovery and higher cell density than conspecifics hosting heterologous Symbiodiniaceae (Gabay et al, 2018, 2019; Medrano et al, 2019; Sproles et al, 2019; Tortorelli et al, 2020). At six months post microalgal symbiont inoculation of the anemones in this study, B1-anemones had a higher cell density than C1-anemones (referred to as "WT10" in Tsang Min Ching et al [2022]). Although samples of this study were collected at 10 months post-inoculation, the higher ceramide levels of

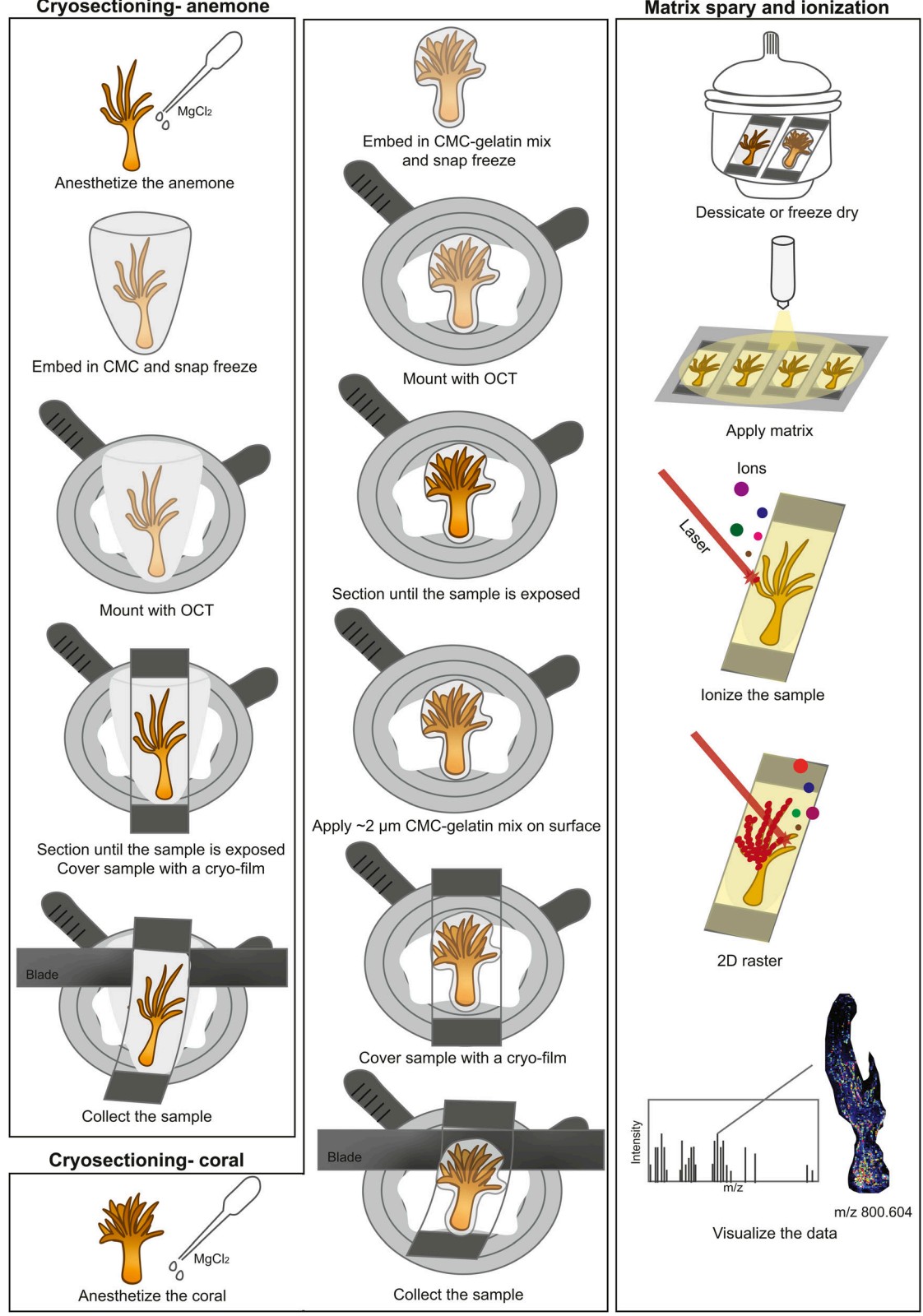

**Figure 1. Spatial metabolomics method.**
Cryosectioning and matrix-assisted laser desorption/ionization mass spectrometry imaging (MALDI-MSI) workflow for the soft-bodied sea anemone *E. diaphana* and the hard-bodied coral *G. fascicularis*. CMC, 2% carboxymethyl cellulose; OCT, optimal cutting temperature compound.

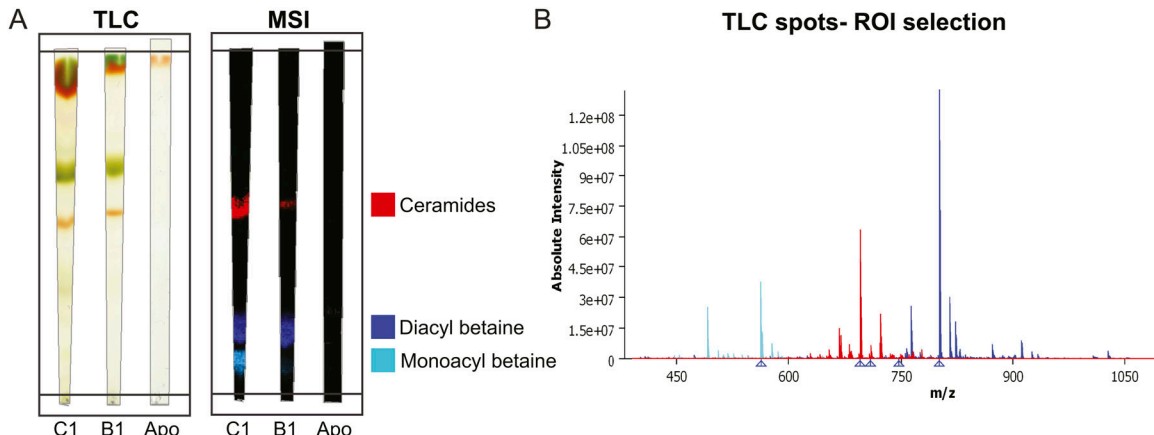

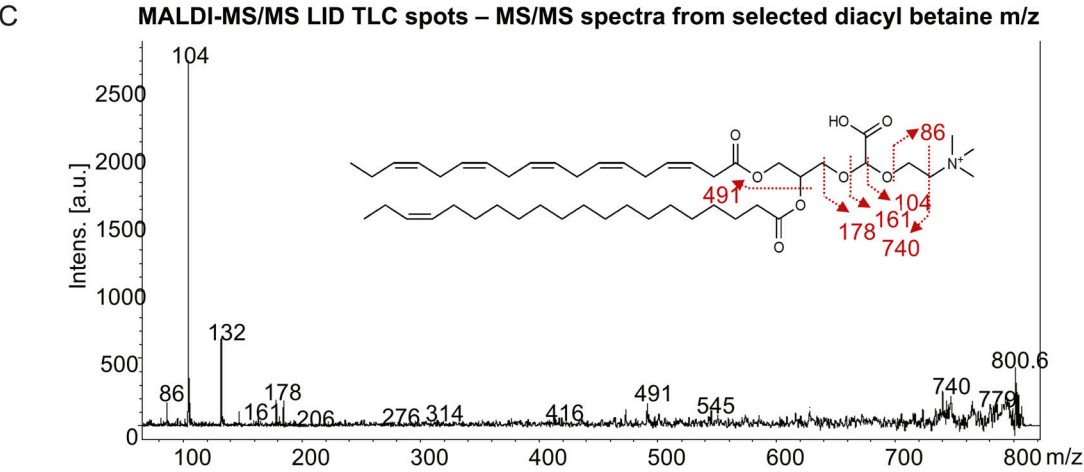

**Figure 2. Example workflow for metabolite annotation.**
**(A)** TLC and mass spectrometry imaging for metabolite annotation on the algal culture *Cladocopium proliferum* (C1) and *Breviolum minutum* (B1), and aposymbiotic anemones (Apo). **(B)** Spectra of three selected metabolite groups (i.e., ceramides, diacyl betaine lipids, monoacyl betaine lipids) based on region-of-interest selection on the TLC. **(C)** Example MS/MS spectra of a diacyl betaine lipid generated using laser induced dissociation.

B1-anemones indicate that B1-anemones may be in a more active symbiont recruitment state than C1-anemones, which is consistent with faster colonization of homologous symbionts observed in the literature. Nevertheless, despite slower symbiont recruitment in C1-anemones, B1- and C1-anemones eventually arrived at the same cell density by ~1.5 yr post-inoculation (Tsang Min Ching et al, 2022).

***Tentacles as the primary site for photosynthesis***
The distribution pattern of betaine lipids suggests that once acquired, microalgal symbionts are mostly transported to the light-exposed anemone tentacles for photosynthate production (Fig 3B and C). We also investigated the spatial distribution of betaine lipids in symbiotic corals and found the same pattern, indicating that this is likely a shared characteristic among cnidarians (Fig 5). Betaine lipids are absent in higher plants but are known to occur in a wide range of marine microalgae (including Symbiodiniaceae [Kato et al, 1996; Roach et al, 2021]), with the three main microalgal betaine lipids being DGTS, diacylglyceryl hydroxymethyl-N,N,N-trimethyl-beta-alanine (DGTA), and diacylglyceryl carboxyhydroxy-methylcholine (DGCC) (Kato et al, 1996; Cañavate et al, 2016). Betaine

lipids were abundant in the Symbiodiniaceae cultures and in symbiotic anemones but were absent in aposymbiotic anemones (Fig 3C and D and Table S3). The only diacylglyceryl-N-trimethylhomoserine (DGTS) betaine lipid detected (m/z 472.363) occurred in high relative intensity in all Symbiodiniaceae cultures but was nearly absent in symbiotic anemones. This suggests that the metabolic state of the microalgal symbionts underwent changes when shifting from the free-living to the symbiotic stage. Conversely, this betaine lipid had high relative intensity in symbiotic corals (Fig 5). Because the microalgal symbiont identity differed between the anemones (*Cladocopium* C1 or *Breviolum* B1) and coral (*Cladocopium* C40), the data reflect that these symbionts may have different lipid constitution or behave differently to each other in the free-living versus symbiotic life stage.

In the sea anemones *Condylactis gigantea* and *Anthopleura elegantissima*, most of the Symbiodiniaceae occur in the tentacles or tentacles and oral disk (Kellogg & Patton, 1983; Dykens & Shick, 1984; Augustine & Muller-Parker, 1998). Consistent with this notion, Dykens and Shick (1984) observed much higher chlorophyll content and oxygen production in the anemone's tentacles and oral disk

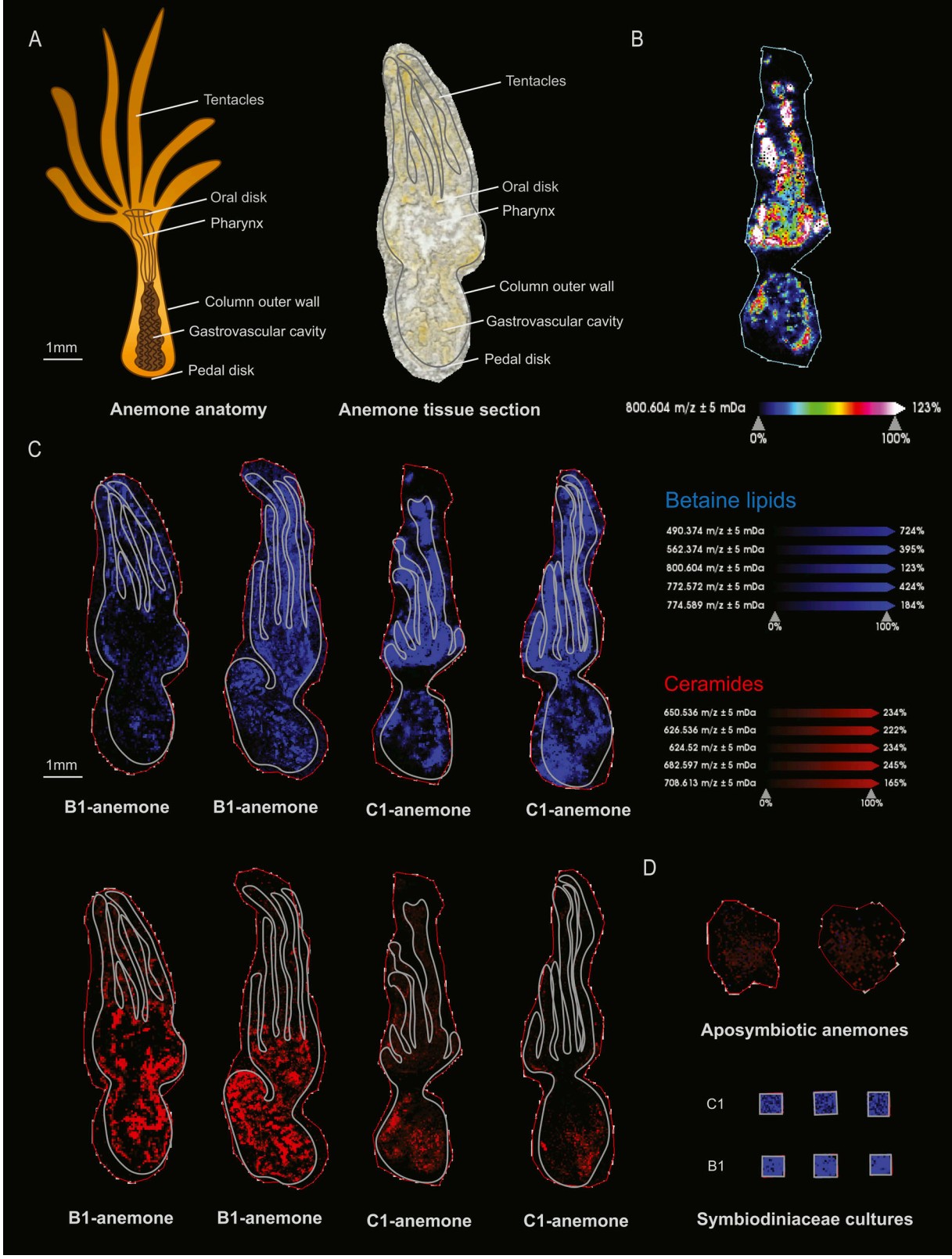

**Figure 3. Spatial metabolite distribution pattern in sea anemones.**
**(A)** Anatomy and a tissue section of the sea anemone *E. diaphana*. **(B)** Spatial intensity of the betaine lipid DGCC m/z 800.604 with a coloured metabolite intensity scale, as an example of the intensity pattern relevant to all other betaine lipids. **(C)** Spatial distribution of five selected betaines lipids (blue, concentrated in the tentacles) versus five selected ceramides (red, concentrated within the tissues surrounding the gastrovascular cavity of symbiotic anemones) in anemones in symbiosis with

than its body column and pedal disc. In *E. diaphana*, whereas no direct observation of Symbiodiniaceae concentration in the tentacles has been previously reported, Symbiodiniaceae colonization in aposymbiotic anemones starts in their oral disk and tentacles, followed by the body column and finally the pedal disk, regardless of the taxonomic identity of the Symbiodiniaceae (Gabay et al, 2018). The higher relative intensity of betaine lipids in the tentacles of *E. diaphana* compared with the body observed here is in line with the spatial distribution pattern of Symbiodiniaceae symbionts in other anemone species. This spatial distribution may reflect preference by the host or microalgae because of the varying *in hospite* light environments, where greater exposure to light in the tentacles is ideal for photosynthesis.

A recent single-cell RNA sequencing study showed unique gene expression patterns in different coral cell types (Levy et al, 2021). Fluorescence-activated cell sorting in the coral *Stylophora pistillata* was able to separate Symbiodiniaceae-hosting gastrodermal cells from non-symbiotic gastrodermal cells; single-cell RNA sequencing revealed Symbiodiniaceae-hosting gastrodermal cells were enriched in 353 host genes including genes related to lipid metabolism, carbonic anhydrases, amino acid, and peptide transporters. In addition, only Symbiodiniaceae-hosting gastrodermal cells expressed all the mRNAs for the galactose-catabolism Leloir pathway, indicating the translocation of galactose from the symbionts to the coral host was occurring within this cell type. As in the present study, these findings allow researchers to link enrichment of a gene product or metabolite to specific cell compartments or structures and make biological inferences on their functionality, highlighting the value of spatial metabolomic and spatial gene expression analysis. The spatial metabolite pattern revealed in MSI studies can also be used to select areas-of-interest to further explore specific research questions. For instance, anemone researchers interested in symbiont regulation and sphingosine rheostat of the host can selectively sample the tissues surrounding the animal's gastrovascular cavity, whereas researchers interested in photophysiology and oxidative stress of the microalgal symbionts can target the animal's tentacles, where focal changes are most likely to occur in the metabolome and the expression of genes.

## Sphingosine rheostat was likely inactive in aposymbiotic anemones

Aposymbiotic anemones were in a different metabolite state compared with symbiotic anemones, with 249 out of 631 metabolites (39.5%) showing statistically significant differences in relative intensity (Figs 3C and D and 6A–D and Table S4). Compared with symbiotic anemones and regardless of their Symbiodiniaceae symbionts' identity, aposymbiotic anemones lacked betaine lipids and had a low relative intensity of ceramides (CER) and diglycerides (DG) in the host tissues (Fig 6D). However, aposymbiotic anemones had a much higher relative intensity of many phosphatidylcholine (PC) lipids in the host component, which dominated the aposymbiotic anemone metabolome (Fig 6D). The Symbiodiniaceae cultures were rich in the 13 major detected chemical species of betaine lipid but

had no or minimal relative intensity in ceramides (Fig 3D). This in combination with the absence of betaine lipids in aposymbiotic anemones (Fig S3 and Table S3) confirms that these lipids are produced by the microalgal symbionts and that the signals detected in the anemone tissues were of microalgal symbiont origin. The low relative intensity in ceramides in aposymbiotic anemones and Symbiodiniaceae cultures (Fig 3D) suggests the high ceramide intensities detected in symbiotic anemones are a consequence of symbiosis, where the presence of symbionts may have triggered de novo synthesis or salvage of sphingolipids in the host that activate the cnidarian-Symbiodiniaceae regulatory system. Alternatively, this regulatory system could be based on a shared metabolic pathway, where the host and symbiont each have parts of the pathway and could only be activated when both parties are present.

## Symbiont identity drives system-level host metabolism spatially

The metabolome of C1-anemones was distinct from that of B1-anemones, with 170 (26.9%) significantly different metabolites (Figs 3C and 6A–D and Table S5). Compared with B1-anemones, C1-anemones had lower relative intensity in many ceramides and fatty acids and several diglycerides (DG), phosphatidylcholines (PC) in the host component (Fig 6D). Of the 26 major ceramide-related chemical species detected, C1-anemones (with heterologous symbionts) had a lower relative intensity in almost all ceramides but higher relative intensity in nearly all the higher molecular weight hexosylceramide (HexCer) and mannosylinositol phosphorylceramide (Fig S2 and Table S2). Of the 13 major chemical species detected for betaine lipids, C1-anemones had significantly higher relative intensity in two metabolites (m/z 772.572, 774.589) compared with B1-anemones (Fig S3 and Table S3). C1-anemones had lower relative intensity in some fatty acids in the host component (Fig 6D). A combined transcriptomic and metabolomic study has demonstrated that *E. diaphana* associated with heterologous microalgal symbionts showed an up-regulation of host innate immunity metabolites and genes, increased lipid catabolism, and decreased transport of fatty acids to the host, whereas those associated with homologous microalgal symbionts showed immunotolerance and symbiont-derived nutrition (Matthews et al, 2017). The lower host relative intensity in fatty acids in C1-anemones (with heterologous symbionts) compared with B1-anemones (with homologous symbionts) is consistent with the above study and in agreement with a previous observation on the same anemones that heterologous symbionts can be slightly less nutritionally beneficial to their host (Tsang Min Ching et al, 2022).

In summary, we have demonstrated a MALDI-MSI workflow that can simultaneously map in situ host and symbiont metabolites in relation to host anatomic structures in soft- and hard-bodied cnidarians. The approach allows researchers to make biological inferences about functionality that would have been overlooked with traditional metabolomics based on bulk tissue analysis, and the methodology is applicable to other marine invertebrates with a delicate soft body or with a hard skeleton and to other symbiotic

*Breviolum minutum* (B1-anemones) and *Cladocopium proliferum* (C1-anemone). **(D)** Relative intensities of betaine lipids (blue) and ceramides (red) in the Symbiodiniaceae cultures and aposymbiotic anemones. Note that aposymbiotic anemones are small compared with symbiotic ones because the absence of microalgal symbionts resulted in reduced nutrient supply and hence anemone growth.

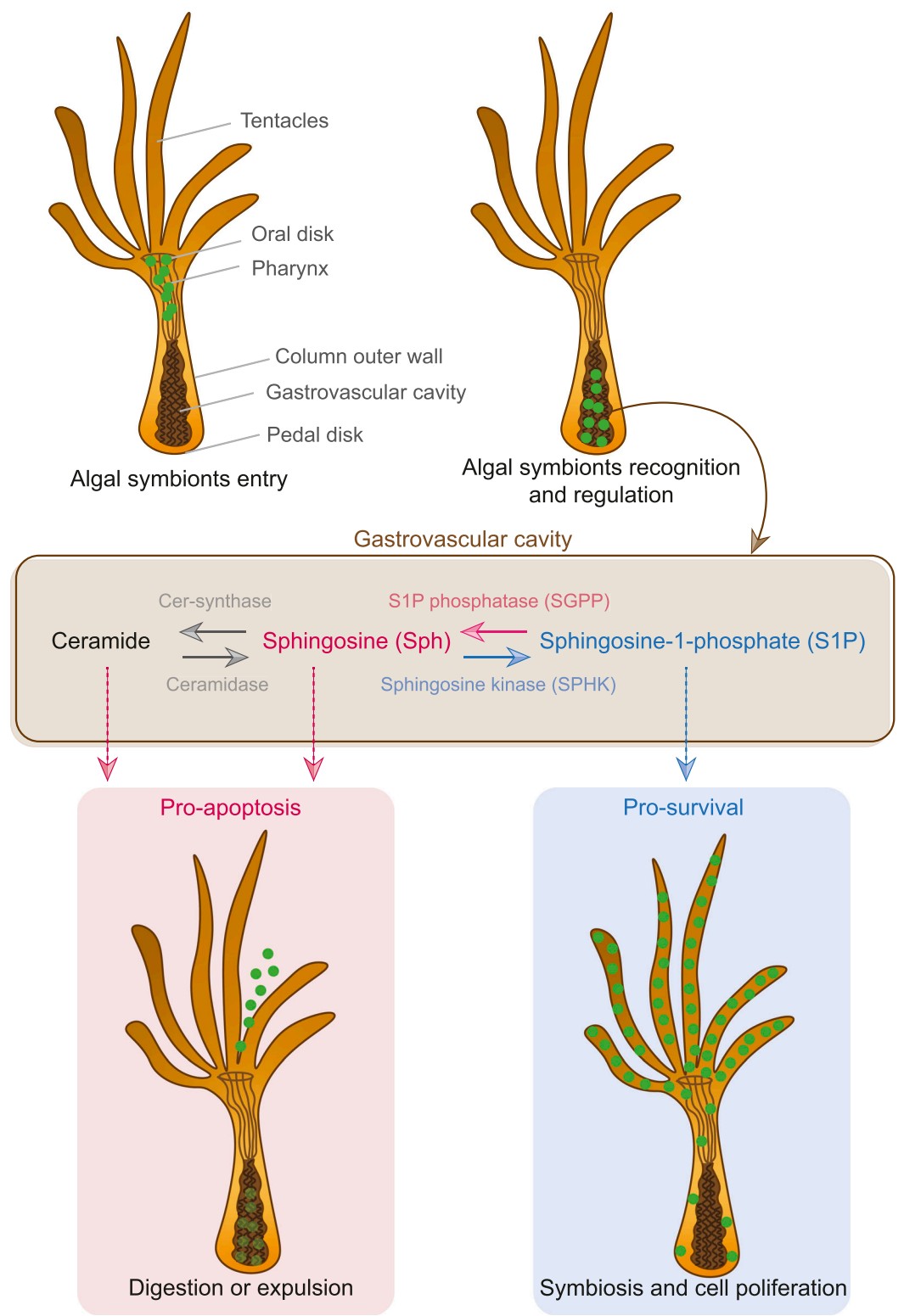

**Figure 4. Cnidarian sphingosine rheostat.**
The mode of entry of microalgal symbionts into a cnidarian host and the potential regulatory role of sphingolipids in the cnidarian-Symbiodiniaceae symbiosis.

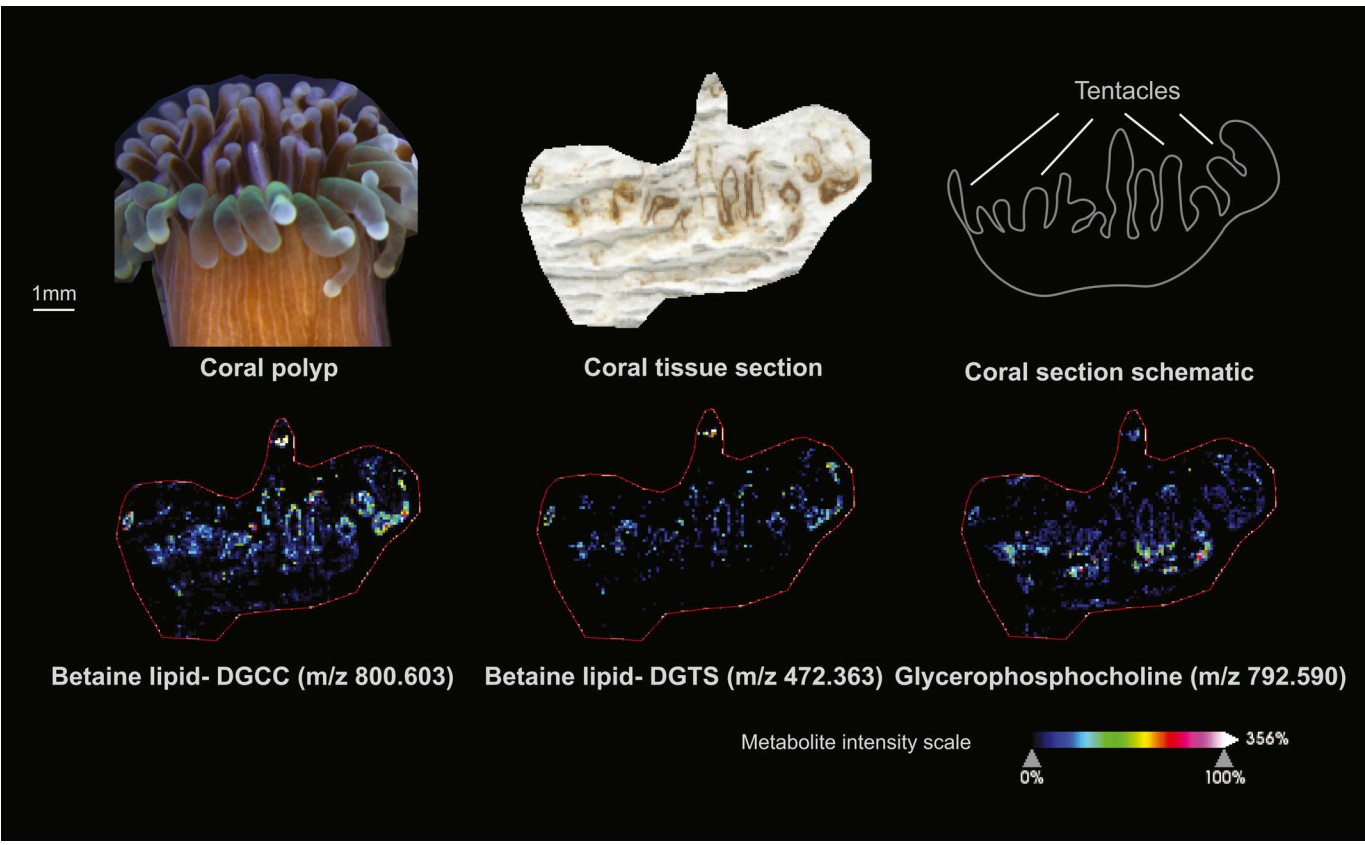

**Figure 5.  Spatial distribution of three selected metabolite in a single polyp of the coral *G. fascicularis*.**
Note the concentration of betaine lipid (microalgal signal) throughout the tentacles of the symbiotic coral and concentration of glycerophosphocholine (PC, host signal) at the base of the coral tentacles.

systems. This is the first study to reveal the cnidarian-Symbiodiniaceae metabolome in situ across the entire anemone and coral polyp. Using the cnidarian-Symbiodiniaceae symbiosis model of the sea anemone *E. diaphana*, MALDI-MSI demonstrated how microalgal symbiont identity can drive system-level change in host metabolism spatially. Questions around the role of a symbiont on host fitness within a given environment are best addressed by sensitive measurements at the specific foci of nutrient exchange. Areas-of-interest identified in MSI can be isolated (e.g., using laser microdissection) for detailed characterization, such as metabarcoding to profile the microbial community and metagenomics to examine functional potential (Wada et al, 2022). With the increasing amount of evidence of a shift in host–microbe dynamics from mutualism to commensalism to parasitism/pathogenicity under a changing climate across ecosystems (Baker et al, 2018; Chagas et al, 2018), novel insights into host–microbe interactions provided by spatial metabolomics are becoming increasingly important.

## Materials and Methods

### Experimental design and sample collection

For the soft-bodied cnidarians, GBR-sourced *E. diaphana* (genotype AIMS4) in symbiosis with the homologous *Breviolum minutum*

(hereafter referred to as B1-anemones, which were inoculated with symbiont culture SCF 127-01, ITS2 profile: B1-B1o-B1p) or the heterologous *Cladocopium proliferum* (Butler et al, 2023) (hereafter referred to as C1-anemones, which were inoculated with symbiont culture SCF 055-01.10, ITS2 profile: C1-C1b-C1c-C42.2-C1br-C1bh-C1cb-C72k) were used (Table S6 and Fig S4). Their symbiont identity was confirmed by ITS2 metabarcoding six months before sampling for MSI (Tsang Min Ching et al, 2022) and reverified three months post-sampling that there was no change (Sakamoto, 2021). *E. diaphana* were sampled 10 mo post-inoculation for MSI and were maintained under ambient temperature of 27°C at 30 $\mu$mol m$^{-2}$ s$^{-1}$ (12:12 h, light: dark). For hard-bodied cnidarians, the coral *G. fascicularis* in symbiosis with *Cladocopium* C40 (ITS2 profile: C40-C3-C115-C40h) was obtained from Palm Islands (Fig S4), GBR and kept under ambient temperature of 27°C and at 130–150 $\mu$mol m$^{-2}$ s$^{-1}$ at full sun (12:12 h light: dark). *E. diaphana* and *G. fascicularis* were fed 5 d a week with freshly hatched *Artemia nauplii* or microalgae mix, respectively. To avoid metabolite contamination from the food source, feeding was stopped five days before sampling. Three samples were collected for each cnidarian group (i.e., B1-anemones, C1-anemones, aposymbiotic anemone, and *G. fascicularis*). Aposymbiotic (i.e., microalgal symbiont free) anemones were produced by a modified menthol bleaching method detailed in the study by Tsang Min Ching et al (Matthews et al, 2016; Tsang Min Ching et al,

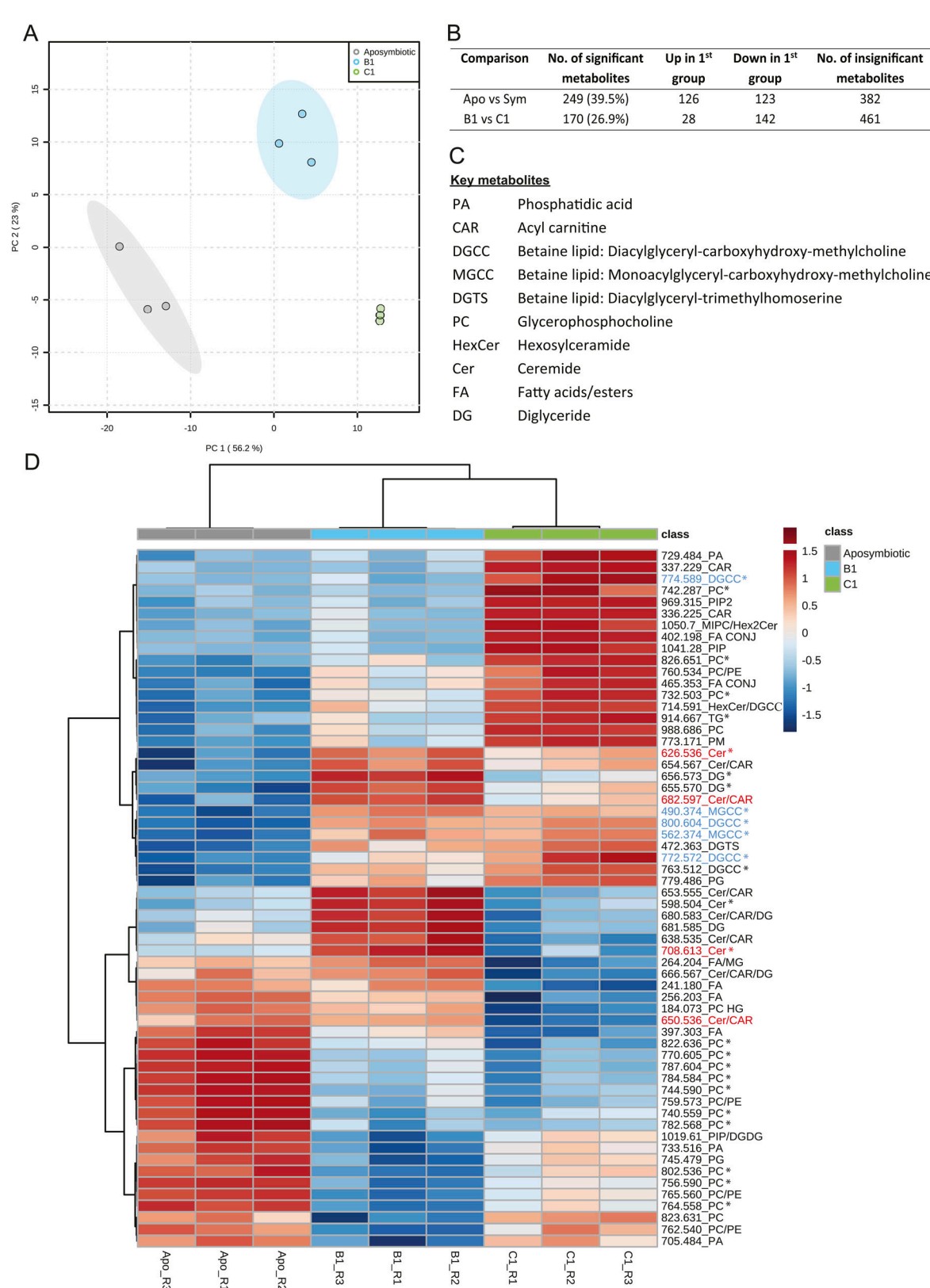

**Figure 6. Statistical comparisons of aposymbiotic, B1- and C1-anemones.**
**(A)** PCA using all 631 metabolites (n = 3 samples per anemone group). Note that the 95% confidence ellipse for the three C1 samples (green) is not visible at the scale presented because of the close proximity of the data points. **(B)** Pairwise comparison results using all 631 metabolites. A metabolite is considered significant when FDR < 0.05 and FC > 2.0. **(C)** (Tables S4 and S5). **(D)** Abbreviation and full names of key metabolites (D) Heatmap of the top 60 significantly different metabolites in an ANOVA

2022). For *G. fascicularis*, single coral polyps were removed with a bone cutter and left to rest in the aquaria for 7 h before fixation (Table 1). The cnidarians were each placed in an individual well of a 12-well plate with 2 ml of seawater and left in the dark for 30 min to allow them to relax and fully extend their tentacles.

To avoid mucus formation (which would hinder MS ionization) and their nematocysts from firing during the sampling process, cnidarians were anaesthetized with 1 ml of 0.4 M MgCl$_2$ and left in the dark for a further 30 min. Anaesthetized cnidarians were then rinsed twice in MQ water and carefully dried with Kimwipes and embedded to maintain the natural shape of the animals for cryosectioning following the methods adapted from Kawamoto & Kawamoto, 2014, 2021; Wada et al, 2016; Boughton et al, 2020. For *E. diaphana*, each sample was placed in an embedding container with 1 ml of embedding media (2% carboxymethyl cellulose, CMC) and snap frozen on dry ice (Fig 1). The CaCO$_3$ skeleton of *G. fascicularis* cryosectioning required optimization; hence, a pilot study was conducted with a range of embedding media (CMC, agar, CMC mixed with gelatin, CMC mixed with agar) at different concentrations to identify the most suitable embedding media. The media that consistently yielded the most intact coral sections (i.e., a ratio of 10 ml 2% CMC: 0.123 g of gelatin) was then used for embedding of the samples to be subjected to MSI. The *G. fascicularis* polyps were dipped into the CMC-gelatin mix with a pair of forceps and snap frozen on dry ice. Cnidarian samples were wrapped in aluminum foil and stored at −80°C until further processing.

### Cryosectioning and matrix spray

Cryosectioning was conducted with a Leica CM 1860 at −20°C (*E. diaphana*) or −25°C (*G. fascicularis*) (Fig 1 and Table 1). Samples were mounted on the cryostat with an optimal cutting temperature compound (OCT) while ensuring that the tissue sections for imaging were not contaminated by OCT. Samples were sectioned at 12 μm thickness until arriving to the middle of the animal, where maximum areas of the frozen body and tentacles were exposed. A low-profile, stainless steel disposable blade was used for the soft-bodied *E. diaphana* and a more robust, high-profile disposable diamond blade was employed for the hard-bodied *G. fascicularis*. A cryofilm was then attached to the exposed sample and a tissue section was collected at 12 μm thickness onto the cryofilm (Fig 1). To minimize scratches on the tissue sections caused by scarping of the CaCO$_3$ skeleton during sectioning in *G. fascicularis*, a thin layer (~2 μm) of the CMC-gelatin mix was applied on the exposed sample surface to strengthen its integrity before collecting a section. Four consecutive sections were collected per sample. For *E. diaphana*, tissue sections were placed in a desiccator for 30 min under a small amount of dry ice, which allowed the sections to gradually come to room temperature during desiccation. For *G. fascicularis*, the desiccator method was insufficient to maintain

tissue integrity, and the tissue sections were instead freeze dried for 12 min under ~−50°C and ~0.08 mbar.

Two desiccated sections per sample were mounted on a stainless steel sheet with carbon tapes and coated with a matrix (A-Cyano-4-hydroxycinnamic acid, HCCA) to assist ionization (Fig 1). For each matrix spray run, 30 mg of HCCA was dissolved in 6 ml of solvent (70% acetonitrile, 30% H$_2$O with 0.1% trifluoroacetic acid) and loaded into the HTX TM-Sprayer. Matrix coating was conducted under 75°C at the flow rate of 100 μl min$^{-1}$ for four passes (*E. diaphana*) or 70 μl min$^{-1}$ for six passes (*G. fascicularis*). Tissue sections were desiccated in a desiccator for a further 20 min before imaging (Table 1). In addition, one desiccated anemone section was imaged with fluorescence lifetime imaging for histological information (Fig S5) and compared with an *E. diaphana* histological study (Lam et al, 2017).

### MALDI-MSI analysis

MALDI-MSI analysis was performed on a Bruker SolariX (7T XR hybrid ESI–MALDI–FT–ICR–MS) with a mass resolving power of 200,000 and equipped with a SmartBeam II UV laser. Before data collection, the instrument was calibrated with a red phosphorus standard to ensure that its mass error was less than 1.5 ppm. Two technical replicates were imaged per sample and two samples were imaged per MSI run. The area-of-interest for imaging was defined using flexImaging 4.1 (Bruker Daltonics) and data acquisition was controlled via Bruker Daltonics ftmsControl 2.1.0. Spectra were collected at a spatial resolution of 50 μm and a range of 150–2,000 m/z under positive ion mode. Laser diameter and power were set to 45 μm and 38% (*E. diaphana*) or 52% (*G. fascicularis*), and a total of 250 (*E. diaphana*) or 500 (*G. fascicularis*) laser shots were applied at each 50 μm pixel at a frequency of 2 kHz.

### Data processing

Raw spectra files (.mis) were uploaded to SCiLS and combined into a single file (.slx) with the intervals set to ± 0.5 mD (Supplemental Datas 2–5). A m/z peak list was generated using the sliding window function and a threshold cutoff of 9,000 was applied, where most of the noises were removed. To remove background peaks (contributed by, e.g., instrument noise, CMC, or gelatin instead of the biological sample), a region-of-interest was created on the background area to identify and remove those associated peaks. A total of 631 peaks remained afterward denoising and each was visualized on SCiLS to confirm that they were associated with the biological samples (Supplemental Data 1). The intensities (average peak area) of the 631 peaks of each technical replicate were exported as excel files, using root mean square to normalize potential intensity differences between MSI runs.

Because each tissue section varied slightly in 2D area, the intensity values were normalized to the total surface area of the

---

between aposymbiotic anemones, anemones in symbiosis with the homologous B1 (*B. minutum*) and anemones in symbiosis with the heterologous C1. Only metabolites with an annotation are shown here. *Indicates that the metabolites were annotated by MALDI-TLC or MS/MS, and metabolites without a * were annotated by library match on METASPACE or Roach et al (2021). See Table S1 for full names of all metabolite abbreviations. Metabolites with colour indicate that their spatial distribution patterns are shown in Fig 3. "R" in the sample name refers the replicate number.

**Table 1.  Sample preparation and MALDI-MSI parameters of the sea anemone *E. diaphana* and the coral *G. fascicularis*.**

|  | Sea anemone *E. diaphana* | Coral *G. fascicularis* |
| --- | --- | --- |
| Sampling | Collect directly with a disposable pipette | Remove with a bone cutter and leave to rest |
| Embedding media | 2% CMC | 2% CMC with gelatin (10 ml CMC: 0.123 g gelatin) |
| Embedding method | Place in an embedding container with 1 ml CMC | Dip sample into the CMC-gelatin mix with a pair of forceps |
| Cryosectioning °C | −20°C | −25°C |
| Cryosectioning blade | Low-profile, stainless steel disposable blade | High-profile, disposable diamond blade |
| Desiccation method | Leave in desiccator for 30 min | Freeze dry for 12 min |
| Matric coating | 100 $\mu$l min$^{-1}$ for four passes | 70 $\mu$l min$^{-1}$ for six passes |
| Laser power | 38% | 52% |
| Laser shot | 250 | 500 |

section to account for size differences. The surface area of a tissue section was identified in SCiLS by overlaying the ion density maps of the most prominent peaks associated with the anemone host (m/z 792.590, glycerophosphocholine) and Symbiodiniaceae (m/z 800.604, betaine lipid DGCC) (Fig S6A–C). The use of these biological peaks accurately identified the tissue area, allowing gaps within a section (e.g., the empty space between two anemone tentacles) to be excluded from the surface area calculation. The ion density maps were exported to ImageJ and converted to 8-bit, and the "biological" surface area was calculated by applying a signal threshold (consistently applied at 30). The intensity of each peak was divided by the surface area of the section for normalization. Technical replicates were then combined and the mean intensity of a peak was used for statistical analyses.

In addition, we also investigated the presence of peptides in anemones and coral samples at m/z > 1,000. Among the 631 peaks, a total of 61 were > 1,000 m/z, all of which were visualized on SCiLS. Previous studies suggest that anemone venoms should be found exclusive in the tentacles (Mitchell et al, 2017; Madio et al, 2018); hence, a peak is considered as a putative venom if it shows such a distribution pattern.

### Metabolite annotation

The peak list (631 peaks) was exported from SCiLS and peaks were initially annotated with LipidMaps, ChEBI, and HMDB databases within METASPACE, with reference to known coral-symbiont metabolites (Roach et al, 2021). Metabolite lists were generated by a high-resolution mass spectrometry (HR-MS) match of the precursor ions. Peak annotations were then further verified focusing on betaine-based lipids, ceramides, pigments, PC-based lipids, and select ions from groups where there was sufficient signal-to-noise (S/N) using MALDI-thin layer chromatography (MALDI-TLC) to add a retardation factor (Rf), in lieu of retention time, and maintain the same ionization mode for analysis (Fig 2A and B). Positional information on bond position and chain length has been left ambiguous.

For TLC analysis, 1 ml of 1 × 10$^6$ cells of the two Symbiodiniaceae cultures and two aposymbiotic anemones were included. Symbiodiniaceae cultures were collected and centrifuged at 1,000 rcf for 5 min to remove the culture media. The pellets were

resuspended in 1 ml MQ water and centrifuged again at 1,000 rcf for 5 min. The supernatant was removed and the pellets were snap frozen on dry ice. Aposymbiotic anemones were rinsed in MQ water and dried with Kimwipes, before being snap frozen on dry ice. For the extraction, 500 $\mu$l of extraction buffer (CHCl$_3$: MeOH, 2: 1) was added to the frozen samples, which were then sonicated and vortexed for 20 min. Fifty $\mu$L of LC–MS-grade water was added to the sample and centrifuged at 1,000 rcf for 5 min. A total of 200 $\mu$l of a sample (the bottom phase) was pipetted to a new Eppendorf tube and dried under N$_2$ flow. Samples were resuspended in 5 $\mu$l of CHCl$_3$, 1 $\mu$l of which was spotted on a TLC plate. The plate was allowed to develop for 30 min in a glass beaker with a mobile phase solvent consisting of 1.4 $\mu$l of LC–MS-grade CHCl$_3$, EtOH, H$_2$O and 0.28 $\mu$l of triethanolamine. The plate was desiccated for 30 min, before being sprayed with HCCA and analyzed with MALDI-MSI with the aforementioned methods (with the exception of spatial resolution, laser diameter, and power, which were set to 200 $\mu$m, 90 $\mu$m, and 60%, respectively). Select metabolites from Rf spots were analyzed using MALDI-MS/MS, laser-induced dissociation to match to LC–MS/MS fragmentation patterns for identification (Fig 2C). MALDI-LID-MS/MS was conducted on a Bruker Ultraflextreme MALDI-TOF, with a tuned LIFT method based on the lower mass range.

### Statistical analysis

Statistical analysis on the various *E. diaphana* groups was performed in MetaboAnalyst 5.0 and the normalized data was log transformed with no scaling applied. Data normality and homogeneity were visually confirmed. All 631 peaks were used to generate PCAs and a one-way ANOVA was used to test for differences among all groups (B1-anemones, C1-anemones, aposymbiotic anemones). The top 60 most significantly different peaks were visualized with heatmaps generated using Euclidean distance and Ward clustering algorithm. The data were subset into pairs of interest (aposymbiotic versus symbiotic anemones; B1-anemones versus C1-anemones) and *t*-tested, with the *P*-values corrected by the Benjamini–Hochberg method (Benjamini & Hochberg, 1995). A peak was considered significant when the $P_{adj}$ < 0.05 and fold change (FC) > 1.3, where FC was calculated based on the non-log–transformed data. Metabolites found to be significant were spatially visualized and confirmed as either a host component

based on their correlation with the host tissue distribution and their presence in the aposymbiotic sample or as an algal component based on correlation with symbiont distribution and detection in the TLC analysis of Symbiodiniaceae cultures. For *G. fascicularis*, the spatial distribution of specific metabolites of interest was visualized in SCiLS, but no statistical comparison was made.

## Data Availability

All data are available in the main text or the supplementary materials. Surface area–normalized metabolite relative intensities are supplied as Supplemental Data 1. Metabolite annotations on MetaSpace are available at https://metaspace2020.eu/ (project name: spatial metabolomics for symbiotic marine invertebrates). Raw spectra files for anemones (Supplemental Data 2 and 3) and corals (Supplemental Data 4 and 5) are available at: Chan et al (2023). Additional information related to this article can be requested from the authors.

## Supplementary Information

## Acknowledgements

We thank SJ Tsang Min Ching and R Sakamoto for anemone supply; the National Sea Simulator team, especially C Thompson and L Koukoumaftsis, for coral collection; Metabolomics Australia staff, especially V Lui and V Narayana, for fruitful discussion and MSI technical support. This research was supported by the Australian Research Council Laureate Fellowship to MJH van Oppen (FL180100036), the Paul G. Allen Family Foundation, and the Reef Restoration and Adaptation Program, which is funded by the partnership between the Australian Governments Reef Trust and the Great Barrier Reef Foundation.

### Author Contributions

WY Chan: conceptualization, data curation, formal analysis, validation, investigation, visualization, methodology, project administration, and writing—original draft, review, and editing.
D Rudd: conceptualization, resources, data curation, software, formal analysis, validation, investigation, visualization, methodology, and writing—review and editing.
MJH van Oppen: conceptualization, resources, supervision, funding acquisition, project administration, and writing—original draft, review, and editing.

### Conflict of Interest Statement

The authors declare that they have no conflict of interest.

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
