## [Reviewer comments · Life Science Alliance]

Spatial metabolomics for symbiotic marine invertebrates

Wing Yan Chan, David Rudd, Madeleine van Oppen
DOI: <https://doi.org/10.26508/lsa.202301900>

Corresponding author(s): Dr. Wing Yan Chan (University of Melbourne; Australian Institute of Marine Science) and Madeleine J.H. Van Oppen

Review Timeline:

Submission Date:	2023-01-04
Editorial Decision:	2023-02-17
Revision Received:	2023-05-08
Editorial Decision:	2023-05-09
Revision Received:	2023-05-09
Accepted:	2023-05-10

Scientific Editor: Eric Sawey

Transaction Report:

No Peer Review Process File is available with this article, as the authors have chosen not to make the review process public in this case.

Re: Life Science Alliance manuscript #LSA-2023-01900-T

Dr Wing Yan Chan
The University of Melbourne, Australian Institute of Marine Science
Australia

Dear Dr. Chan,

Thank you for submitting your manuscript entitled "Spatial metabolomics for symbiotic marine invertebrates" to Life Science Alliance. The manuscript was assessed by expert reviewers, whose comments are appended to this letter. We invite you to submit a revised manuscript addressing the Reviewer comments.

Thank you for this interesting contribution to Life Science Alliance. We are looking forward to receiving your revised manuscript.

Sincerely,

Eric Sawey, PhD
Executive Editor
Life Science Alliance
<http://www.lsa-journal.org>

- A letter addressing the reviewers' comments point by point.
- An editable version of the final text (.DOC or .DOCX) is needed for copyediting (no PDFs).
- High-resolution figure, supplementary figure and video files uploaded as individual files: See our detailed guidelines for preparing your production-ready images, <https://www.life-science-alliance.org/authors>
- Summary blurb (enter in submission system): A short text summarizing in a single sentence the study (max. 200 characters including spaces). This text is used in conjunction with the titles of papers, hence should be informative and complementary to the title and running title. It should describe the context and significance of the findings for a general readership; it should be written in the present tense and refer to the work in the third person. Author names should not be mentioned.
- By submitting a revision, you attest that you are aware of our payment policies found here: <https://www.life-science-alliance.org/copyright-license-fee>

B. MANUSCRIPT ORGANIZATION AND FORMATTING:

RE: Life Science Alliance Manuscript #LSA-2023-01900-TR

Dr. Wing Yan Chan
University of Melbourne
School of Biosciences
BioSciences 2, Building 122 The University of Melbourne
Parkville, VIC 3010
Australia

Dear Dr. Chan,

Thank you for submitting your revised manuscript entitled "Spatial metabolomics for symbiotic marine invertebrates". We would be happy to publish your paper in Life Science Alliance pending final revisions necessary to meet our formatting guidelines.

- please upload your supplementary figures as single files and add a separate figure legend section (with your main and supplementary figures) to your main manuscript text
- please add ORCID ID for secondary corresponding author-they should have received instructions on how to do so
- please incorporate the supplementary methods into the main materials and methods section
- please make sure that table files are uploaded as a doc or excel file
- please add a figure callout for figure 6A-C to your main manuscript text

A. FINAL FILES:

B. MANUSCRIPT ORGANIZATION AND FORMATTING:

Sincerely,

RE: Life Science Alliance Manuscript #LSA-2023-01900-TRR

Dr. Wing Yan Chan
University of Melbourne
School of Biosciences
BioSciences 2, Building 122 The University of Melbourne
Parkville, VIC 3010
Australia

Dear Dr. Chan,

Thank you for submitting your Methods entitled "Spatial metabolomics for symbiotic marine invertebrates". It is a pleasure to let you know that your manuscript is now accepted for publication in Life Science Alliance. Congratulations on this interesting work.

DISTRIBUTION OF MATERIALS:

Again, congratulations on a very nice paper. I hope you found the review process to be constructive and are pleased with how the manuscript was handled editorially. We look forward to future exciting submissions from your lab.

Sincerely,
